# Molecular surveillance of zoonotic pathogens from wild rodents in the Republic of Korea

**Kyoung-Seong Choi**[1]*, **Sunwoo Hwang**[1], **Myung Cheol Kim**[2], **Hyung-Chul Cho**[1], **Yu-Jin Park**[1], **Min-Jeong Ji**[1], **Sun-Woo Han**[3], **Joon-Seok Chae**[3]

1 Department of Animal Science and Biotechnology, College of Ecology and Environmental Science, Kyungpook National University, Sangju, Republic of Korea, 2 Department of Ecological Science, College of Ecology and Environmental Science, Kyungpook National University, Sangju, Republic of Korea, 3 Laboratory of Veterinary Internal Medicine, BK21 FOUR Future Veterinary Medicine Leading Education and Research Centre, Research Institute for Veterinary Science and College of Veterinary Medicine, Seoul National University, Seoul, Republic of Korea

* kschoi3@knu.ac.kr

## Abstract

### Background

Rodents are recognized as major reservoirs of numerous zoonotic pathogens and are involved in the transmission and maintenance of infectious diseases. Furthermore, despite their importance, diseases transmitted by rodents have been neglected. To date, there have been limited epidemiological studies on rodents, and information regarding their involvement in infectious diseases in the Republic of Korea (ROK) is still scarce.

### Methodology/Principal findings

We investigated rodent-borne pathogens using nested PCR/RT-PCR from 156 rodents including 151 *Apodemus agrarius* and 5 *Rattus norvegicus* from 27 regions in eight provinces across the ROK between March 2019 and November 2020. Spleen, kidney, and blood samples were used to detect *Anaplasma phagocytophilum*, *Bartonella* spp., *Borrelia burgdorferi* sensu lato group, *Coxiella burnetii*, *Leptospira interrogans*, and severe fever with thrombocytopenia syndrome virus (SFTSV). Of the 156 rodents, 73 (46.8%) were infected with *Bartonella* spp., 25 (16.0%) with *C. burnetii*, 24 (15.4%) with *L. interrogans*, 21 (13.5%) with *A. phagocytophilum*, 9 (5.8%) with SFTSV, and 5 (3.2%) with *Borrelia afzelii*. Co-infections with two and three pathogens were detected in 33 (21.1%) and 11 rodents (7.1%), respectively. *A. phagocytophilum* was detected in all regions, showing a widespread occurrence in the ROK. The infection rates of *Bartonella* spp. were 83.3% for *B. grahamii* and 16.7% for *B. taylorii*.

### Conclusions/Significance

To the best of our knowledge, this is the first report of *C. burnetii* and SFTSV infections in rodents in the ROK. This study also provides the first description of various rodent-borne pathogens through an extensive epidemiological survey in the ROK. These results suggest that rodents harbor various pathogens that pose a potential threat to public health in the ROK. Our findings provide useful information on the occurrence and distribution of zoonotic

**Data Availability Statement:** All data generated during this study are included in the article and

supplementary material. Raw sequencing files were deposited and are available in the repository, WEB LINK for A. phagocytophilum (https://www.ncbi.nlm.nih.gov/nuccore/OR287077.1, OR287091.1), for B. grahamii (https://www.ncbi.nlm.nih.gov/nuccore/OR288176.1, OR288190.1), for B. taylorii (https://www.ncbi.nlm.nih.gov/nuccore/OR288191.1, OR288193.1), for B. afzelii (https://www.ncbi.nlm.nih.gov/nuccore/OR284310.1, OR284311.1), for C. burnetii (https://www.ncbi.nlm.nih.gov/nuccore/OR284312.1, OR284321.1), for L. interrogans (https://www.ncbi.nlm.nih.gov/nuccore/OR284322.1, OR284324.1), and for SFTSV (https://www.ncbi.nlm.nih.gov/nuccore/OR257718.1, OR257726.1).

**Funding:** The author(s) received no specific funding for this work.

**Competing interests:** The authors have declared that no competing interests exist.

pathogens disseminated among rodents and emphasize the urgent need for rapid diagnosis, prevention, and control strategies for these zoonotic diseases.

## Author summary

Rodents live almost everywhere in the world, adapt to extremely diverse habitats, and transmit various infectious diseases to humans and other animals. All six pathogens were detected in rodents. Our findings demonstrated that 66.7% (104/156) of rodents were infected with at least one pathogen. We also observed differences in the pathogens detected in rodents by province. These results provide evidence that rodents play an important role in the transmission of SFTSV. Although we did not screen for all rodent-borne diseases, these data provide information about emerging rodent-borne diseases disseminated in the ROK and emphasize the risk of occurrence of rodent-borne diseases.

## Introduction

Rodents are globally abundant and well-known reservoirs and vectors of infectious diseases that affect both livestock and humans [1, 2]. The current global change context (e.g., land-use change, urbanization, and temperature increase) is particularly suitable for the expansion of several rodent species beyond their natural distribution areas [3,4]. Rodents are widespread in rural and urban areas and cause numerous human infections in regions where humans are in close contact with them. Rodents are reservoir hosts for at least 60 zoonotic diseases and play a vital role in disease transmission by spreading disease directly through contact or bites or indirectly through arthropods or food contamination [5–7]. Despite their potential threat to public health, there has not been much focus on diseases transmitted by rodents [8,9]. Moreover, the control of rodents is difficult due to their behavioral plasticity, life history traits, and high breeding potential [3].

*Anaplasma phagocytophilum* is a tick-transmitted, obligatory intracellular zoonotic bacterium that infects the neutrophils of various hosts, including humans, dogs, cats, horses, domestic, and wild animals [10–13]. The clinical signs of *A. phagocytophilum* infection range from asymptomatic to serious symptoms of veterinary and public health importance. The occurrence of *A. phagocytophilum* infection is increasing along with climate change worldwide. A broad variety of animal species are known to harbor *A. phagocytophilum*, and humans are incidental dead-end hosts [14]. Vertebrate hosts are crucial for the maintenance and circulation of this pathogen in enzootic foci. In particular, small rodents and wild ruminants have been suggested as primary reservoirs [15–19]. In the United States, the white-footed mouse (*Peromyscus leucopus*) is considered a well-established reservoir species [20,21]. In the Republic of Korea (ROK), *A. phagocytophilum* has also been detected in small mammals such as rodents and shrews (*Crocidura lasiura*) [22,23].

*Bartonella* spp. are facultative intracellular bacteria that cause persistent infections in the erythrocytes and endothelial cells of mammalian hosts [24]. The clinical manifestations caused by these species are characterized by fever, endocarditis, myocarditis, neuroretinitis, lymphadenopathy, and a range of vascular pathologies [24–28]. More than 30 *Bartonella* spp. and three subspecies are currently recognized [29], and at least 20 species are associated with rodents, indicating that rodents serve as potential reservoirs for zoonotic *Bartonella* spp. [30–32]. Among the rodent-adapted *Bartonella* spp., *B. elizabethae*, *B. grahamii*, *B. rochalimae*, *B. tribocorum*, *B. vinsonii*, and *B. washoensis* have been found to cause human infections [32,33]. In general, *Bartonella* spp. have been considered to be transmitted by arthropods [24,31].

Although *Bartonella* infections are widely distributed in rodents in different geographic regions [34–41], there is very little information on the distribution and prevalence of these species in rodents in the ROK [22,42,43].

Lyme borreliosis (LB) is one of the most common vector-borne diseases in North America and Eurasia and is caused by a spirochete belonging to the *Borrelia burgdorferi* sensu lato (s.l.) group [44]. Among this group, *B. burgdorferi* sensu stricto (s.s.), *B. afzelii*, and *B. garinii* are the major causative agents of LB in humans and exhibit different geographical distributions [45,46]. These species are transmitted between vertebrate hosts and tick vectors [47]. *B. burgdorferi* s.s. occurs in North America and Europe and has various reservoir hosts (e.g., rodents and birds), whereas *B. afzelii* and *B. garinii* occur in Eurasia and can only use specific vertebrate hosts, namely, rodents and birds, respectively [44,45]. Different *Borrelia* species cause different symptoms in humans. For instance, *B. burgdorferi* s.s. infection is associated with Lyme arthritis, whereas *B. garinii* is mostly linked to neuroborreliosis, and *B. afzelii* infection is related to a chronic skin condition known as acrodermatitis [44,48–50]. In the ROK, *B. burgdorferi* s.l. was first detected in 1993 and has been sporadically identified in ticks, dogs, horses, wild rodents, and humans [51–56].

*Coxiella burnetii* is an obligate intracellular bacterium with a worldwide distribution and is the causative agent of Q fever in humans and a wide range of animals [57]. It is highly infectious and has the ability to form spore-like particles that can withstand harsh environmental conditions and can be easily dispersed by airflow [58]. Humans acquire *C. burnetii* infection through inhalation of contaminated aerosols or dust particles [59]. Q fever is a public health concern as it ranks as one of the 13 leading global priority zoonoses. Moreover, it has been considered a potential biological weapon due to its widespread availability, aerosolized spread, and environmental stability [60]. The clinical manifestations of *C. burnetii* infection include fever and flu-like symptoms. The major sources of these infections are infected ruminants, which experience issues of abortion and infertility. Ticks and rodents are also known as natural reservoirs of *C. burnetii* [61]. Other studies have recently performed molecular characterization of this pathogen in domestic animals in the ROK [57,62]; however, these studies had a limited spatial distribution and were species-specific.

Leptospirosis is a zoonotic infectious disease with a global distribution and is caused by a spirochete of the genus *Leptospira* [63,64]. It infects more than one million people annually, with 60,000 deaths recorded [65]. *Leptospira* is maintained in several wild and domestic animal hosts through renal carriage and is excreted in the urine for several months [66,67]. Infection in humans and animals primarily occurs through direct contact with the urine of infected hosts or indirect exposure to contaminated water, soil, or food [68]. Its clinical manifestations in humans range from a mild febrile illness to life-threatening renal failure, pulmonary hemorrhage, and/or cardiac complications [69]. Recent studies have suggested that an increase in the incidence of leptospirosis in humans is often associated with heavy rainfall and flooding [70,71]. Rodents are considered the most important reservoir of pathogenic *Leptospira* spp. and contribute to disease transmission because of their close contact with humans and domestic animals [72]. *L. interrogans*, *L. borgpetersenii*, and *L. kirschneri* are the most abundant species circulating in humans and animals worldwide [73], with *L. interrogans* being the most commonly described in rodents [72].

Severe fever with thrombocytopenia syndrome (SFTS) is an emerging tick-borne viral disease that has been primarily reported in China, the ROK, Japan, Vietnam, and Taiwan [74–78]. SFTS is formally caused by *Bandavirus dabieense* [also commonly known as SFTS virus (SFTSV)], which belongs to the genus *Bandavirus* in the family *Phenuiviridae*. SFTSV infections are characterized by high fever, fatigue, myalgia, gastrointestinal symptoms, thrombocytopenia, and multiorgan failure [74,79]. SFTSV can also spread from person to person through

exposure to the blood of an infected individual [80]. Due to its potential as a public health threat, SFTS was chosen as one of nine emerging diseases given a priority for research by the World Health Organization in 2017 [81]. As humans are often in close contact with domestic animals and may encounter rodents when they work outdoors, transmission between animals and humans is another possible major transmission route [82]. The overall mortality rate of this disease has been reported to be 3–30% in different countries [74,83,84]. Although SFTSV has been identified in various animals, its natural reservoir hosts have not been determined.

As demonstrated in the abovementioned studies, rodents may be involved in the transmission cycles of various diseases. Recently, the incidences of several infectious diseases have been rapidly increasing worldwide due to climate warming. Rodent populations are also growing exponentially due to climate change and urbanization. To date, most studies on rodent-borne diseases in the ROK have been primarily focused on identifying hantavirus infections. Although rodents are considered important reservoirs of zoonotic infectious pathogens, epidemiological information regarding their involvement in infectious diseases is limited in the ROK. Therefore, the aims of this study were to investigate the occurrence of some rodent-borne diseases, characterize their genetic relationships, and determine the roles of rodents as reservoir hosts for these diseases.

## Methods

### Ethical statement

Rodent collection was approved by the Seoul National University Institutional Animal Care and Use Committee (No. SNU-190524-2-1) and performed according to the Seoul National University Guidelines on the care and use of laboratory animals.

### Sample collection

Rodents were captured using Sherman traps ($3 \times 3.5 \times 9$-inch folding traps; H.B. Sherman Traps, Tallahassee, FL, USA) from 27 regions in eight provinces across the country between March 2019 and November 2020. Rodents were captured from at least one region in each province (S1 Table). The traps were set in locations where human infections with SFTSV had been reported based on statistical data from the Korea Disease Control and Prevention Agency. To capture rodents, rural (agricultural land such as regions with rivers, valleys, mountains, and lakes) and peri-urban (human residential and farm) areas were selected. Rural areas were defined as regions with natural landscapes and minimal urban impact, whereas peri-urban areas were defined as transitional zones between urban and rural areas. A total of 60 traps were installed in each capture region in lines in 3-m intervals between 5 p.m. and 6 p.m. and retrieved the next day between 9 a.m. and 10 a.m. after setup: 60 traps per night were set in each region. The capture duration for every site was 14–15 h, and each capture site was sampled only once. The captured rodents [rural ($n = 127$) and peri-urban ($n = 29$)] were transported to the laboratory in an icebox with the traps, morphologically identified, and euthanized using $CO_2$. Thereafter, blood, spleen, and kidney samples were collected from each animal. A whole blood sample was also collected in a serum separation tube, and the serum was separated and used for RNA extraction.

### DNA/RNA extraction and PCR analysis

DNA was extracted from spleen (10 mg) and kidney (25 mg) samples using the DNeasy Blood & Tissue Kit (Qiagen, Hilden, Germany) according to the manufacturer's instructions and stored at −20°C until analysis. Splenic DNA was subjected to PCR amplification to detect *A.*

**Table 1. Primer information used for PCR analysis.**

| Pathogen | Target genes | Sequences (5′–3′) | Sizes (bp) | Annealing temp./Time |
|---|---|---|---|---|
| *Anaplasma phagocytophilum* | 16S rRNA | TCCTGGCTCAGAACGAACGCTGGCGGC | 1,433 | 50˚C/30 s |
| | | AGTCACTGACCCAACCTTAAATGGCTG | | |
| | | GTCGAACGGATTATTTTTATAGCTTGC | 926 | 56˚C/30 s |
| | | CCCTTCCGTTAAGAAGGATCTAATCTCC | | |
| *Bartonella* spp. | ITS | TTCAGATGATGATCCCAAGC | 639 | 55˚C/30 s |
| | | AACATGTCTGAATATATCTTC | | |
| | | CCGGAGGGCTTGTAGCTCAG | 499 | 55˚C/30 s |
| | | CACAATTTCAATAGAAC | | |
| *Borrelia* spp. | ospA | GGGAATAGGTCTAATATTAGCC | 665 | 42˚C/60 s |
| | | CACTAATTGTTAAAGTGGAAGT | | |
| | | GCAAAATGTTAGCAGCCTTGAT | 392 | 56˚C/60 s |
| | | CTGTGTATTCAAGTCTGGC | | |
| *Coxiella burnetii* | IS1111 | TATGTATCCACCGTAGCCAGTC | 687 | 54˚C/30 s |
| | | CCCAACAACAACCTCCTTATTC | | |
| | | GAGCGAACCATTGGTATCG | 203 | 54˚C/30 s |
| | | CTTTAACAGCGCTTGAACGT | | |
| *Leptospira interrogans* | rpoB | GTTCCAACATGCAACGYCAR | 1,649 | 52˚C/60 s |
| | | GTTGAAGGATTCRGGRATAC | | |
| | | TYATGCCKTGGGAAGGWTAC | 1,023 | 56˚C/30 s |
| | | GCATRTCRTCKGACTTGATG | | |
| SFTSV | S | CATCATTGTCTTTGCCCTGA | 461 | 52˚C/40 s |
| | | AGAAGACAGAGTTCACAGCA | | |
| | | AAYAAGATCGTCAAGGCATCA | 346 | 55˚C/40 s |
| | | TAGTCTTGGTGAAGGCATCTT | | |

*SFTSV: severe fever with thrombocytopenia syndrome virus

*phagocytophilum*, *Bartonella* spp., *Borrelia* spp., and *C. burnetii*, and kidney DNA was subjected to PCR to detect *L. interrogans*. These pathogens were screened using each specific primer with the nested PCR method under the following conditions: 93–95˚C for 5 min, followed by 30–40 cycles of 93–95˚C for 1 min, the annealing temperature of each pathogen, 72˚C for 1 min, and a final extension step at 72˚C for 10 min (Table 1). Distilled water was used as a negative control in all PCR analyses. Secondary PCR products were visualized on 1.5% agarose gels stained with ethidium bromide.

RNA was extracted from 200-μL aliquots of serum using the Gene-spin Viral DNA/RNA Extraction Kit (iNtRON Biotechnology, Seongnam, ROK) according to the manufacturer's instructions. The viral RNA was stored at − 80˚C until use. Each RNA sample was tested using nested reverse transcription-polymerase chain reaction (RT-PCR) assays to detect the small (S) segment of SFTSV. Primary PCR was performed using one-step RT-PCR premix (Solgent, Daejeon, ROK) under the following conditions: initial step of 30 min at 50˚C and 15 min at 95˚C for denaturation, followed by 40 cycles of 20 s at 95˚C, 40 s at 52˚C, and 30 s at 72˚C, with a final extension step of 5 min at 72˚C. Nested PCR was conducted using 1 μL of the primary PCR product as a template (BIOFACT, Daejeon, ROK). The reaction for the nested PCR consisted of 25 cycles of 20 s at 94˚C, 40 s at 55˚C, and 30 s at 72˚C. The primer information used to detect SFTSV is listed in Table 1. Secondary PCR products were visualized on 1.5% agarose gels stained with ethidium bromide.

## Phylogenetic analysis

The secondary PCR products were purified using the AccuPrep PCR Purification Kit (Bioneer, Daejeon, ROK) according to the manufacturer's instructions and directly sequenced (Macrogen Inc., Seoul, ROK). Less than five PCR-positive samples from each pathogen were utilized for all sequencing analyses. In cases in which there were more than five positive samples, we selected only a few samples and used them for the sequencing analysis. All the nucleotide sequences obtained for each pathogen were aligned using the BioEdit software and then compared with reference sequences from the National Center for Biotechnology Information database (http://www.ncbi.nlm.nih.gov) to determine their similarity using Geneious Prime 2022.2 software (http://www.geneious.com). Phylogenetic analysis of each pathogen was performed using the maximum-likelihood method implemented in MEGA11 using the best substitution model. Bootstrap values were calculated by analyzing 1,000 replicates to evaluate the reliability of clusters. The models used in this study were K2 + G for *A. phagocytophilum*, Tamura 3-parameter + G + I for *Bartonella* spp., Tamura-Nei for *Borrelia* spp., and the Kimura 2-parameter model for *C. burnetii*, *L. interrogans*, and SFTSV. The nucleotide sequences obtained in this study were assigned the following accession numbers: OR287077-OR287091 for *A. phagocytophilum*, OR288176-OR288190 for *B. grahamii*, OR288191-OR288193 for *B. taylorii*, OR284310-OR284311 for *B. afzelii*, OR284312-OR284321 for *C. burnetii*, OR284322-OR284324 for *L. interrogans*, and OR257718-OR257726 for SFTSV.

## Statistical analysis

The infection rates were calculated with 95% confidence intervals (CIs). The PCR results for each rodent sample were recorded as negative or positive and were categorized as a single infection or a co-infection with two or three pathogens. Statistical analysis was performed using the SPSS 29.0 software package for Windows (SPSS Inc., Chicago, IL, USA). The association between sex and the infection rate for each pathogen was determined using the chi-square test. A *P*-value $\leq$ 0.05 was considered statistically significant.

# Results

## Sample collection

A total of 175 rodents were captured and morphologically classified as follows: 151 *Apodemus agrarius* (striped field mouse) (70 males and 81 females), 5 *Rattus norvegicus* (Norway rat) (2 males and 3 females), and 19 unknown. *A. agrarius* was found in most of the capture sites in the ROK, whereas *R. norvegicus* was captured only in two provinces (Table 2). Unknown samples were excluded from this study, and the remaining 156 rodents were used for data analysis.

## Prevalence of pathogens detected in the captured rodents

The presence of six pathogens was investigated by PCR analysis from the two species, *A. agrarius* and *R. norvegicus*. Of the 156 rodents, 104 (66.7%; 95% CI: 59.3–74.1) were infected with at least one pathogen. The infection rate was 64.3% in females (54/84) and 66.6% in males (48/72). There was no significant difference between single infection or co-infections and sex (*P* = 0.886). The infection rates of each pathogen by sex are shown in Table 3. In all the six pathogens, no significant difference was found in the infection rate between the sexes (Table 3).

In terms of pathogens, *Bartonella* spp. were frequently detected (46.8%; 95% CI: 39.0–54.6), followed by *C. burnetii* (16.0%; 95% CI: 10.2–21.8), *L. interrogans* (15.4%; 95% CI: 9.7–21.1), *A. phagocytophilum* (13.5%; 95% CI: 8.1–18.9), SFTSV (5.8%; 95% CI: 2.1–9.5), and *Borrelia*

**Table 2. Number of rodents captured by province.**

| Province | *Apodemus agrarius* | *Rattus norvegicus* | Total |
|---|---|---|---|
| Gyeonggi | 12 | - | 12 |
| Gangwon | 18 | - | 18 |
| Chungbuk | 19 | - | 19 |
| Chungnam/Daejeon | 4 | - | 4 |
| Jeonbuk | 13 | - | 13 |
| Jeonnam | 4 | - | 4 |
| Gyeongbuk | 76 | 4 | 80 |
| Gyeongnam | 5 | 1 | 6 |
| Total | 151 | 5 | 156 |

"-": no rodents captured

spp. (3.2%; 95% CI: 0.4–6.0) (Table 4). The details of the pathogens identified by province are shown in Table 4. All six pathogens were detected in Gangwon, Chungbuk, and Gyeongbuk provinces. Five of the six pathogens, excluding SFTSV, were found in Gyeongnam province, whereas only one pathogen was detected in Chungnam and Jeonnam provinces (Table 4). Co-infections with two and three pathogens from the captured rodents were also detected in 33 (21.2%; 95% CI: 14.8–27.6) and 11 (7.1%; 95% CI: 3.1–11.1) animals, respectively (Table 5), and co-infections with *Bartonella* spp. and *L. interrogans* were the most frequently detected (Table 5). Information regarding the pathogens identified by region is shown in the map in (Fig 1).

### Phylogenetic trees of rodent-associated pathogens

*Anaplasma phagocytophilum.* *A. phagocytophilum* was detected in rodents from all the examined provinces, indicating that this pathogen is spread throughout the ROK. Of 21 positive samples, 15 were sequenced and confirmed to be *A. phagocytophilum* by a phylogenetic tree analysis based on the 16S rRNA gene (Fig 2). Our sequences exhibited 97.6–99.9% identity with each other and 95.6–100% identity with those reported in the ROK. These 15 sequences were similar to those previously reported from several different hosts such as cattle, dogs, horses, humans, ticks, and rodents in other countries, sharing 96.9–100% nucleotide identities with these. Furthermore, several variants co-existed in the same geographical area. According to the phylogenetic tree, *A. phagocytophilum* was divided into clades 1 and 2, and all our

**Table 3. Association between each pathogen and the sex of the captured rodents.**

| Pathogens | Male (*n* = 72) | Female (*n* = 84) | $\chi^2$ (*P*-value) |
|---|---|---|---|
| *A. phagocytophilum* | 9 | 12 | 0.008 (0.928) |
| *Bartonella* spp. | 40 | 33 | 3.494 (0.062) |
| *Borrelia* spp. | 2 | 3 | 0.000 (1.000) |
| *C. burnetii* | 9 | 16 | 0.796 (0.372) |
| *L. interrogans* | 12 | 12 | 0.035 (0.851) |
| SFTSV | 4 | 5 | 0.000 (1.000) |
| Total | 48 | 54 | 0.020 (0.886) |

*$P < 0.05$

**$P < 0.005$

**Table 4. Number of positive samples in which pathogens were identified in captured rodents according to each province.**

| Province | *A. phagocytophilum* | *Bartonella* spp. | *Borrelia* spp. | *C. burnetii* | *L. interrogans* | SFTSV |
|---|---|---|---|---|---|---|
| Gyeonggi (*n* = 12) | 1 | 5 | - | - | 2 | 2 |
| Gangwon (*n* = 18 | 4 | 12 | 1 | 5 | 6 | 1 |
| Chungbuk (*n* = 19) | 3 | 10 | 1 | 2 | 6 | 2 |
| Chungnam/Daejeon (*n* = 4) | 1 | - | - | - | - | - |
| Jeonbuk (*n* = 13) | 3 | 8 | - | 3 | - | - |
| Jeonnam (*n* = 4) | 1 | - | - | - | - | - |
| Gyeongbuk (*n* = 76) | 7 | 32 | 2 | 14 | 6 | 4 |
| Gyeongnam (*n* = 5) | 1 | 6 | 1 | 1 | 4 | - |
| Total (*n* = 151) | 21 (13.5%) | 73 (46.8%) | 5 (3.2%) | 25 (16.0%) | 24 (15.4%) | 9 (5.8%) |

"-": not detected.

sequences from *A. agrarius* belonged to clade 1 (Fig 2). The difference in sequences between clades 1 and 2 revealed 94.7–98.5% nucleotide identities. Clade 2 had 10 nucleotide differences compared with clade 1. Genetic variants were detected in *A. phagocytophilum* circulating in the ROK.

**Bartonella spp..** *Bartonella* spp. were frequently detected in *A. agrarius* in the ROK, but they were not found in all provinces (Table 4). *Bartonella* spp. were detected in both *A. agrarius* and *R. norvegicus*. Of 73 internal transcribed spacer (ITS) PCR-positive samples, 18 sequences were included in a phylogenetic analysis. Two species of *Bartonella* spp. were identified circulating in the examined rodents using the phylogenetic tree based on the *ITS* gene: *B. grahamii* and *B. taylorii* (Fig 3). The prevalence of *B. grahamii* and *B. taylorii* was 83.3% (15/18) and 16.7% (3/18), respectively. The 15 sequences belonging to *B. grahamii* showed 94.9–100% identity with each other and formed the same group as *B. grahamii* found in leeches (KX270012) and *A. agrarius* (JN810851) reported in the ROK, exhibiting 95.9–99.8% identity with those sequences. Furthermore, another sequence (JN810855) reported from *A. agrarius* in the ROK demonstrated 87.1–90.8% similarity to the sequences reported in our study. The three sequences classified as *B. taylorii* exhibited 100% identity with each other and shared 92.5–100% identity with those belonging to this species.

**Borrelia spp..** *Borrelia* spp. were found in four provinces (Table 4), and the infection rate of *Borrelia* spp. was the lowest (3.2%) compared with those of the other pathogens identified. Among the five PCR-positive samples, only two sequences were obtained and demonstrated 98.6% identity with each other. A phylogenetic analysis based on the outer surface protein A (*ospA*) gene revealed that our sequences were assigned to *B. afzelii* (Fig 4). The two sequences exhibited 98.9–100% homology with those identified previously in *A. agrarius* in the ROK. Our sequences showed 97.8–100% identity with those belonging to this group. Furthermore, these sequences displayed 98.2–99.6% similarity to those reported in humans from Austria, Germany, the Czech Republic, the ROK, and Sweden.

**Coxiella burnetii.** *C. burnetii* was the second most frequently detected pathogen and was identified in both *A. agrarius* and *R. norvegicus*. However, it was found in five different provinces. Of 25 positive samples, 10 sequences were included in a phylogenetic tree based on the *IS1111* gene. These sequences showed 97.5–100% identity with each other. Only one sequence (OR284314) had the closest genetic relationship with strains identified in febrile and pneumonic patients (KP645188 and JF970260), which are known as virulent strains, exhibiting 100% homology with those (Fig 5). The others formed a separate branch, exhibiting 99.0–99.5% identity with two human isolates (KP645188 and JF970260). The phylogenetic tree revealed the presence of genetic variations within the *C. burnetii* sequences.

**Table 5. Co-infections of two or three pathogens detected in captured rodents.**

| Pathogens | No. of positive samples |
|---|---|
| *A. phagocytophilum* + *Bartonella* spp. | 7 |
| *A. phagocytophilum* + *Borrelia* spp. | 1 |
| *A. phagocytophilum* + *C. burnetii* | 1 |
| *Bartonella* spp. + *Borrelia* spp. | 2 |
| *Bartonella* spp. + *C. burnetii* | 7 |
| *Bartonella* spp. + *L. interrogans* | 10 |
| *Bartonella* spp. + SFTSV | 3 |
| *C. burnetii* + *L. interrogans* | 2 |
| *A. phagocytophilum* + *Bartonella* spp.+ *C. burnetii* | 1 |
| *A. phagocytophilum* + *Bartonella* spp.+ *L. interrogans* | 3 |
| *A. phagocytophilum* + *Bartonella* spp.+ SFTSV | 2 |
| *A. phagocytophilum* + *C. burnetii* + *L. interrogans* | 1 |
| *Bartonella* spp. + *Borrelia* spp. + *C. burnetii* | 1 |
| *Bartonella* spp. + *C. burnetii* + *L. interrogans* | 1 |
| *Bartonella* spp. + *L. interrogans* + SFTSV | 2 |

*Leptospira interrogans.* *L. interrogans* was the third most commonly detected pathogen and was also found in both *A. agrarius* and *R. norvegicus*. Of 24 positive samples, only three sequences were obtained, and they had 97.7–99.5% identity with each other. A phylogenetic tree based on the RNA polymerase subunit beta (*rpoB*) gene revealed that these sequences belonged to *L. interrogans* (Fig 6). Two sequences (OR284322 and OR284323) were classified into *L. interrogans* serovar *Lai* and showed 99.2–100% identity with those reported in China and 99.4–100% identity with those reported in *A. agrarius* in the ROK. The other sequence (OR284324) belonged to *L. interrogans* serovar *Manilae*, which has detected in *Mus musculus* in Japan, exhibiting 98.2% similarity (Fig 6). At least two serovars of *L. interrogans* were found to be circulating in *A. agrarius* in the ROK.

**Severe fever with thrombocytopenia syndrome virus.** SFTSV was detected in nine *A. agrarius* samples (5.8%) and found in four different provinces (Table 4). Of the nine SFTSV infections, a single infection with SFTSV was detected only in two *A. agrarius* samples, and the remaining samples were primarily co-infected with other pathogens such as *Bartonella* spp. and *L. interrogans* (Table 5). Nine sequences were obtained and included in the phylogenetic tree. These sequences demonstrated 95.95–100.0% identity with each other. The phylogenetic analysis based on the S segments revealed that five and four sequences were classified into sub-genotype B-2 and genotype D, respectively (Fig 7). The sequences belonging to genotype B-2 exhibited 94.5–97.4% homology with those identified in human and other animal samples in the ROK, whereas four sequences showed 99.7–100.0% identity with those identified in human samples. These results revealed that genotype B-2 is prevalent in the ROK and that genetic variants exist within genotype B-2.

## Discussion

This study demonstrated the infection rate and genetic characterization of zoonotic pathogens by molecular analysis in rodents captured from throughout the ROK. Rodents were trapped from rural and peri-urban areas with frequent movement of people, which may be associated with a high probability of disease transmission because humans and rodents share the same space. In the present study, all six pathogens were detected in rodents. The results demonstrated that 66.7% (104/151) of rodents were infected with at least one pathogen. According to

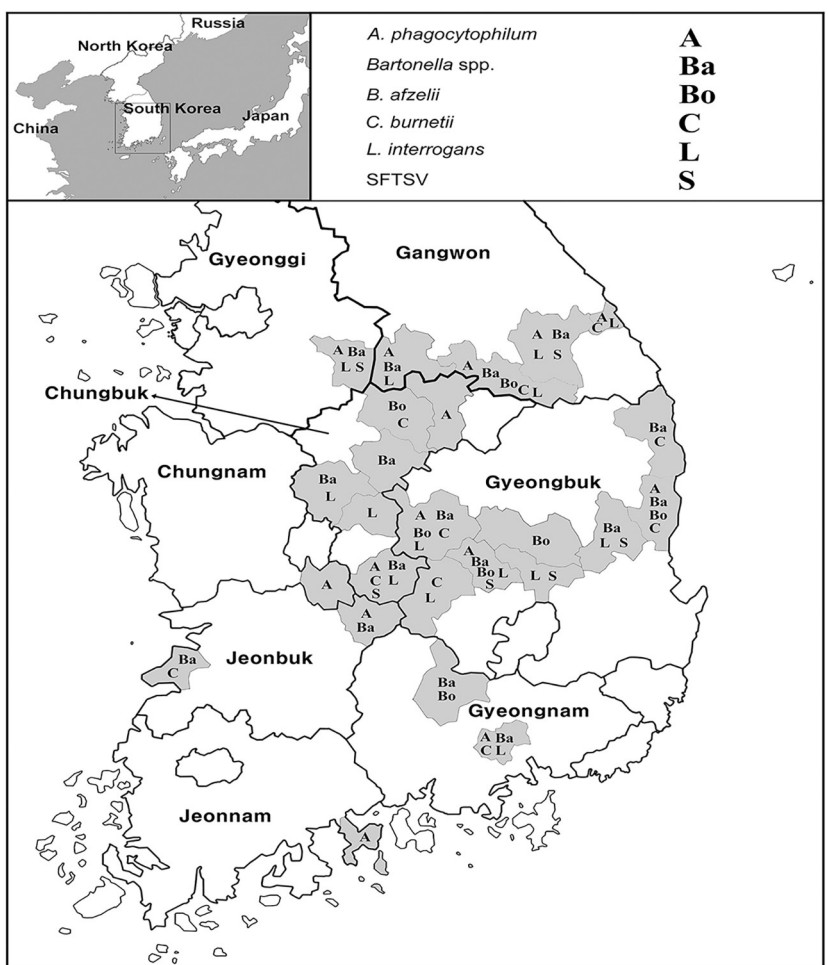

**Fig 1. Maps showing the regions where rodent-borne pathogens were detected in the Republic of Korea.** The word symbol is indicated differently according to each pathogen. Maps were created using NGII [https://www.data.go.kr/data/15062309/fileData.do] and are Korea Open Government License Type 1, which can be freely used without any permission.

our findings, *Bartonella* spp. were most frequently detected, and *Borrelia* spp. were the least commonly detected in rodents. Moreover, by sex, there was no difference in the infection rate and no statistically significant differences between single infection and co-infections. A limitation of this study is that the number of captured rodents in each province varied. Nevertheless, five pathogens were identified in Gyeongnam province, which had the second lowest number of rodents captured. In contrast, although more rodents were captured in Gyeonggi and Jeonbuk provinces than in Gyeongnam province, fewer pathogens were detected. These results imply that increased sample numbers do not necessarily correlate with the probability of pathogen detection. Moreover, because most rodents were captured in rural areas, it is impossible to compare the infection rates between rural and peri-urban areas. At this time, we cannot conclude differences in pathogen detection, but this finding may be influenced by the sampling site where the rodents were captured. Although the infection rate was not very high, *A. phagocytophilum* was found in all provinces. Considering that the number of rodents captured varied by province and that few rodents were captured in some provinces, *A. phagocytophilum* may be the most widespread pathogen in the ROK. Furthermore, to the best of our knowledge,

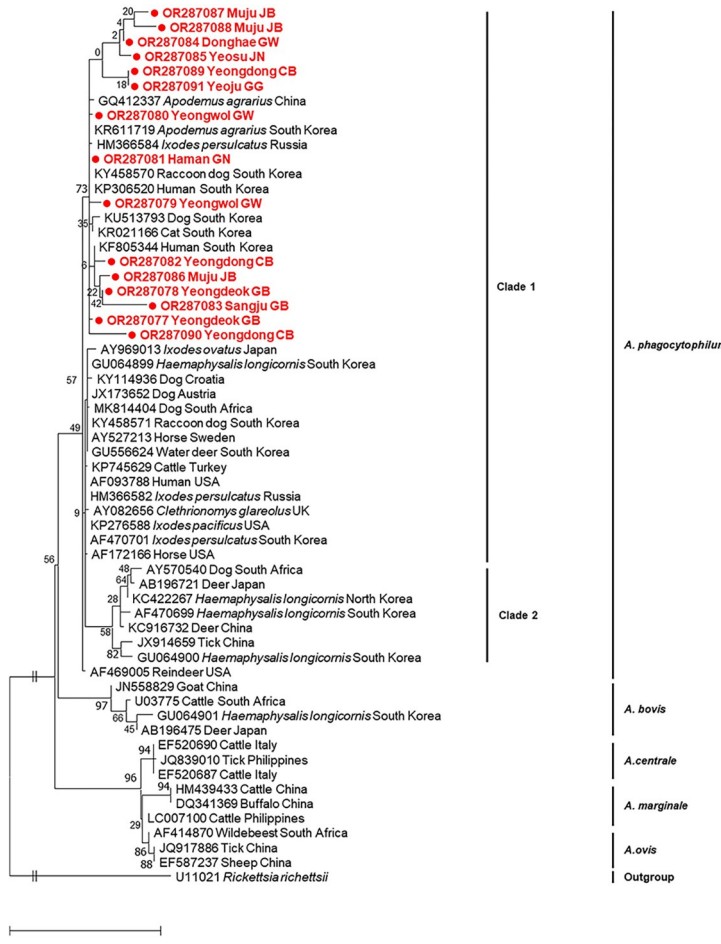

**Fig 2. Phylogenetic tree inferred by maximum-likelihood analysis using the K2 + G model of the 16S rRNA gene sequence of *Anaplasma phagocytophilum*.** The numbers at the nodes are bootstrap values expressed as a percentage of 1,000 replicates. The scale bar indicates nucleotide substitution per site. Samples sequenced from *Apodemus agrarius* are shown in filled circles.

this is the first study to report *C. burnetii* and SFTSV infections in rodents in the ROK and to conduct an extensive study to investigate infections with several pathogens. These data provide valuable information for evaluating the potential risk of rodents in public health.

In this study, all six of the evaluated pathogens were detected using the nested PCR method. In general, nested PCR is more sensitive than qPCR but is also prone to contamination and is more cumbersome [85]. There were the discrepancies between PCRs and sequencing results. We cannot address the reason why the sequencing results were not good at this time. This may be the need for more amounts of genetic material to enable proper sequencing. Nonetheless, we considered PCR-positives to be truly positive, regardless of whether the amplicons could be sequenced. It seems difficult to consider them negative only because some amplicons could not be sequenced.

*Anaplasma phagocytophilum* is known as the third most common tick-borne pathogen in the USA and Europe [86] and has been detected in 20 different rodent species [87]. *A. phagocytophilum* infection rates range considerably in rodent species [87]; this may be explained by differences in small mammals that maintain tick species. In this study, the prevalence of

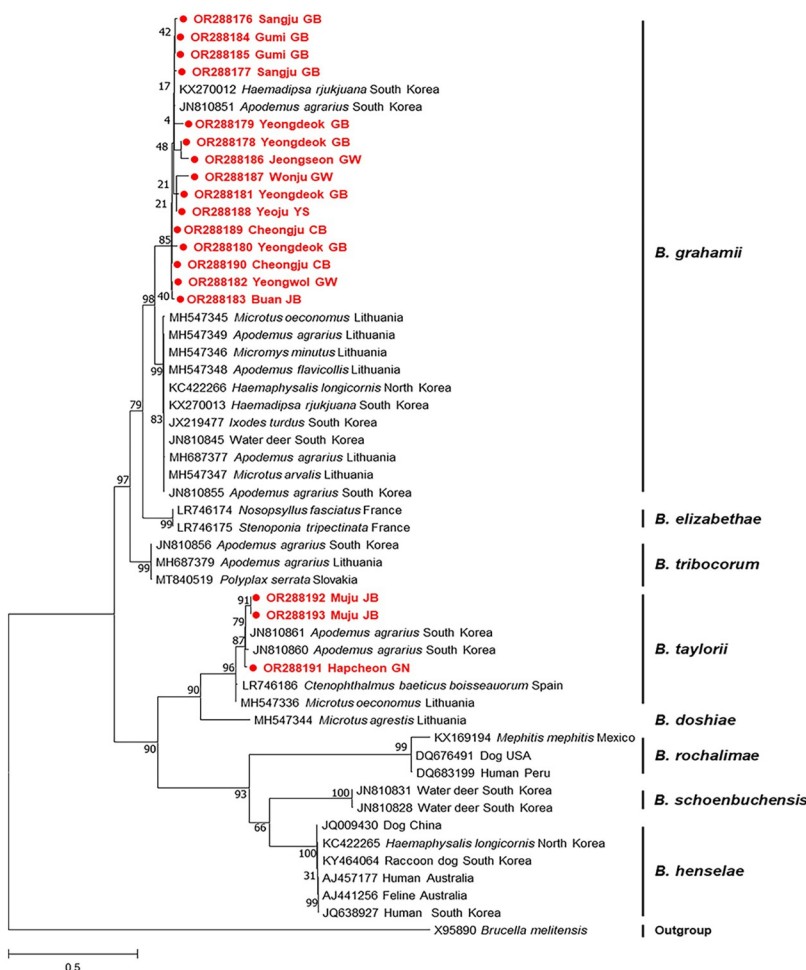

**Fig 3. Phylogenetic analysis based on the *ITS* region of *Bartonella* spp. (maximum-likelihood analysis using the Tamura 3-parameter + G + I model with 1,000 replicates).** The scale bar indicates nucleotide substitution per site. Sequences determined from *A. agrarius* are indicated in filled circles.

*A. phagocytophilum* infections in *A. agrarius* was 13.5%, which was rather lower than reported in a previous study conducted in the ROK (19.1%) [88]. To date, there has been no report of *A. phagocytophilum* infection in *Rattus* spp. in the ROK; however, a high infection rate (31.5%) of *A. phagocytophilum* has been reported in *Rattus* spp. from China [89]. *A. phagocytophilum* has been detected in a variety of animals, including ticks in the ROK, but its pathogenicity still remains unclear. The sequences obtained from rodents showed 97.9–100% identity to *A. agrarius* (KR611719) reported in the ROK. According to the phylogenetic analysis, there are several genetic variants among *A. phagocytophilum* circulating in the ROK. *Haemaphysalis longicornis*, which is found primarily in the ROK, tends to use *A. agrarius* as the major host to maintain *A. phagocytophilum* [23,89], indicating that *A. agrarius* is an enzootic reservoir. Thus, further studies are required to determine the pathogenicity of *A. phagocytophilum* variants circulating in the ROK.

The overall prevalence of *Bartonella* spp. in rodents was 46.8%, which was the highest of the prevalence rates of all the other pathogens examined in this study. However, the detection rate in the present study was lower than that described in a previous report (62.0%) based on the *ITS* gene [43]; this difference may be because of the location where the rodents were captured.

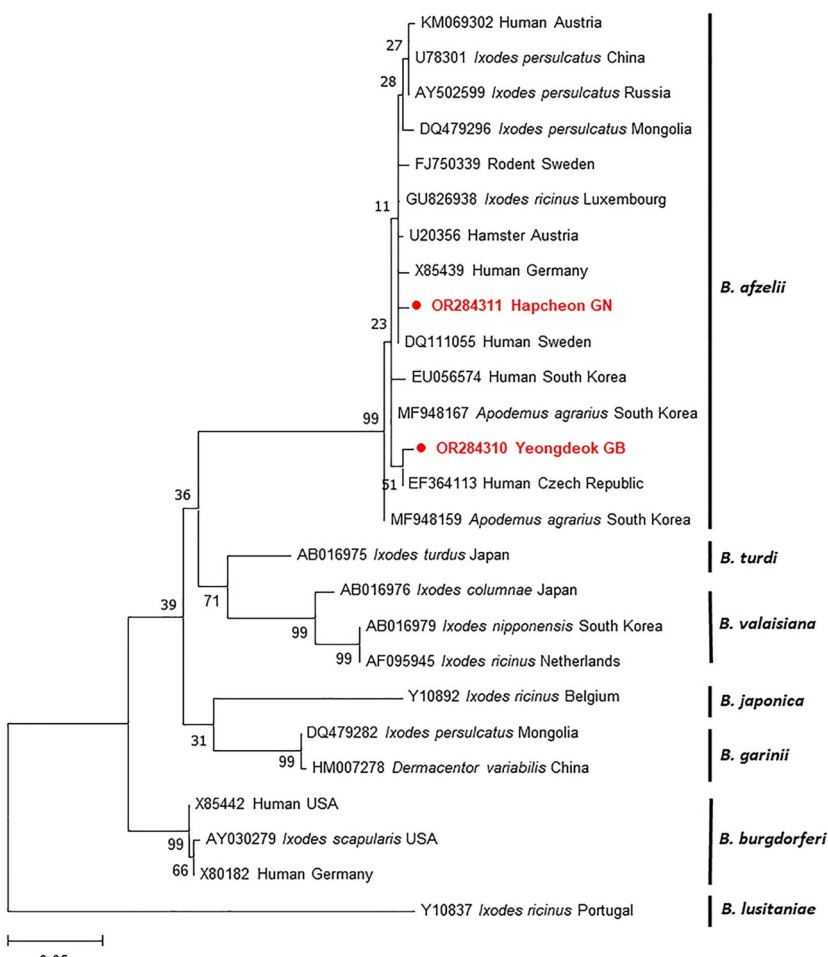

**Fig 4. Maximum-likelihood phylogenetic tree using the Tamura-Nei model based on the *ospA* gene of *Borrelia* spp. Bootstrap values were calculated with 1,000 replicates of the alignment.** The scale bar indicates nucleotide substitution per site. Sequences obtained from *A. agrarius* are symbolized in filled circles.

Moreover, the prevalence of *Bartonella* spp. in rodents varies across countries, e.g., 5.5% in Turkey [37], 23.7% in Lithuania [90], 36.3% in Chile [32], 40.4% in Slovenia [34], and 65.8% in Eastern Germany [41]. In addition, the *Bartonella* spp. that are prevalent in each country are different [27,32,34,38,90–93]. According to the phylogenetic tree, *B. grahamii* has two distinct groups, and the sequences obtained in this study formed the same group and diverged from some sequences previously identified in the ROK. This indicates that genetic variations of *B. grahamii* exist in the ROK. *R. norvegicus* and *R. rattus* are known as major reservoirs for *Bartonella* spp. in several countries [27,94–96], but *Bartonella* spp. have not been detected in other rodent species or in *R. norvegicus* in the ROK [22]. The analysis of *R. norvegicus* in this study was unfortunately limited because only five animals were captured, and their data were excluded. Further studies are necessary to investigate *Bartonella* spp. infection in *R. norvegicus*. The present results demonstrated that *B. grahamii* was the most predominant species, and *B. taylorii* was found in three rodents, which is consistent with the findings of a previous study [43]. *B. grahamii* is a zoonotic pathogen that is associated with neuroretinitis and retinal artery occlusion in humans [25]. *B. taylorii* can cause infection in animals [90], but its pathogenicity remains unclear. In Europe, *B. taylorii* is the predominant species found in rodents [24,37]. In

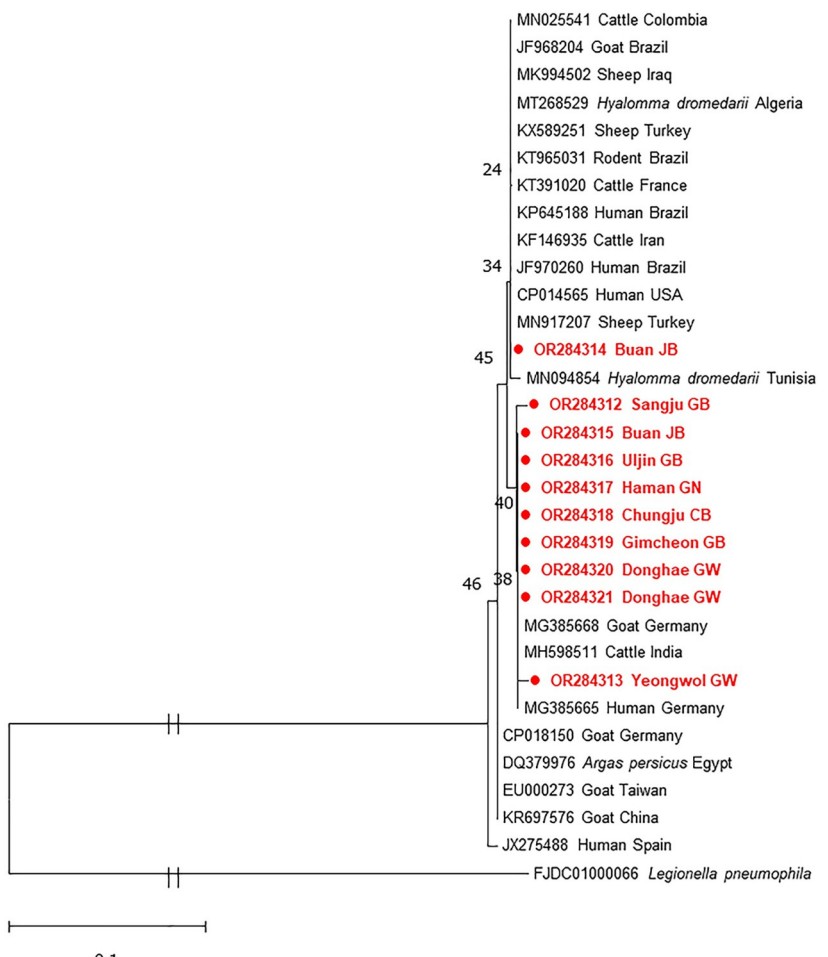

**Fig 5. Maximum-likelihood phylogenetic tree from the *IS1111* gene of *Coxiella burnetii*.** The evolutionary analysis was inferred using the Kimura 2-parameter model. Bootstrap values (1,000 replicates) are indicated in each node. The scale bar indicates nucleotide substitution per site. Sequences determined from *A. agrarius* are highlighted in filled circles.

this study, the *Bartonella* spp. detected were different by province, and in Jeonbuk province, both *B. grahamii* and *B. taylorii* were identified, indicating that the pathogens present in each province are different. Although *B. taylorii* has been detected in some *A. agrarius* in the ROK, its transmission route remains unknown. Considering its high infection rate in *A. agrarius*, potential vectors of this pathogen should be identified to prevent infection.

The detection rate of *Borrelia* spp. in *A. agrarius* was 3.2% and was the lowest infection rate compared with those of the other pathogens examined in this study. Our result was different from those of previous studies that evaluated heart samples from *A. agrarius* (29.6%) [56] and ticks (33.6%) collected from wild rodents in the ROK [97]. This finding can be explained by the differences in the genes and samples used. For instance, Kim et al [56] reported that *B. burgdorferi* s.s. and *B. garinii* infected the spleen and that *B. afzelii* exhibited a high detection rate in the heart; however, *B. burgdorferi* s.s. and *B. garinii* were not detected in the spleen. It is speculated that the number of positive samples was small and could not be detected. Among the *Borrelia burgdorferi* s.l. group, only *B. afzelii* was identified in *A. agrarius*, which supports previous findings that *B. afzelii* is the predominant species in the ROK [54,97]. Furthermore,

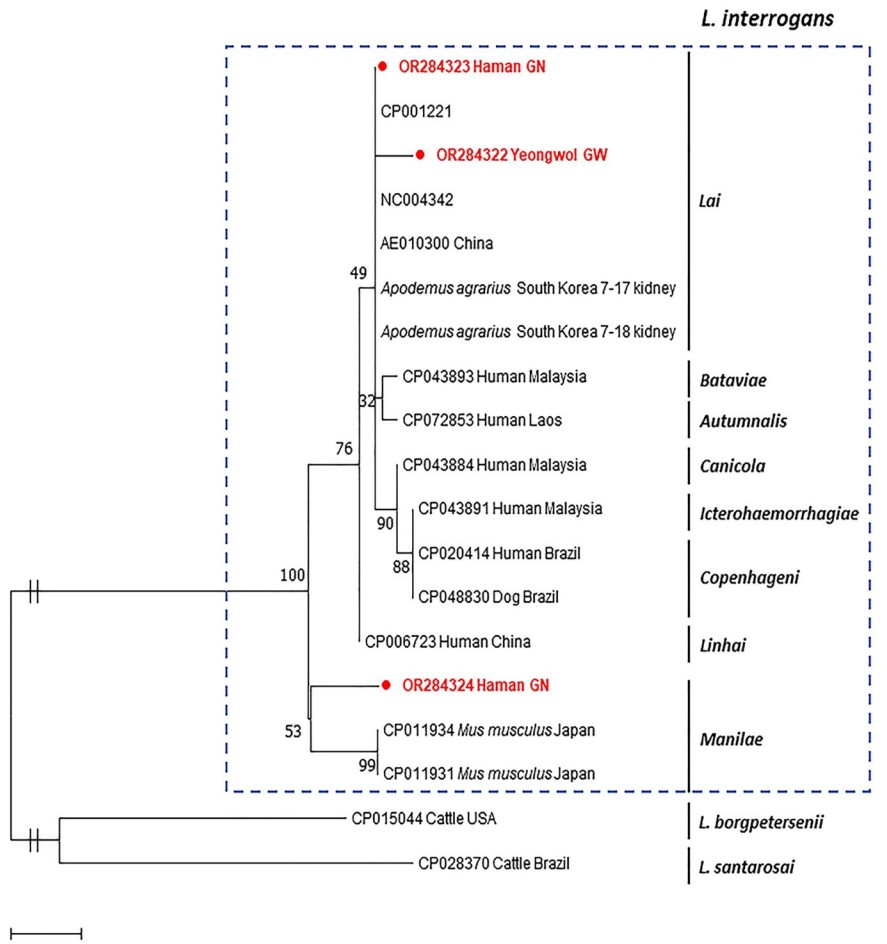

**Fig 6. Phylogenetic analysis based on the *rpoB* gene of *Leptospira interrogans*.** The tree was inferred in MEGA X using maximum-likelihood and Kimura 2-parameter with 1,000 replicates. The scale bar implies nucleotide substitution per site. The box of dash lines indicates *L. interrogans*. Sequences obtained from *A. agrarius* are shown in filled circles.

our results were significantly lower than those reported in rodents in other countries: 24% in Austria [98], 16% in the Czech Republic [99], and 6.3% in Spain [100]. These differences in prevalence may be due to the tick vectors; the common tick vectors of *Borrelia* spp. in the ROK are *Ixodes persulcatus*, *I. nipponensis*, and *I. granulatus* [101]. *B. afzelii* is transmitted by *I. ricinus* and hosted by small mammals, and it is the most common causative agent of human LB [45,102]. The sequences from rodents showed 98.2–99.6% identity with those identified in ticks and humans in several countries. In the ROK, *B. afzelii* has been primarily reported in ticks [54,97,103] and rarely in humans [104]. Nevertheless, information on *B. afzelii* is still lacking. Although the infection rate of *B. afzelii* in rodents was the lowest, our findings suggest that *B. afzelii* may act as a causative agent of LB in the ROK.

This is the first report of *C. burnetii* in *A. agrarius* in the ROK. In this study, *C. burnetii* exhibited the second highest infection rate (16.0%). Nevertheless, our results were lower than those reported in China (18%) [105], Senegal (22.4%) [106], and Zambia (45%) [107] but higher than those reported in Brazil (4.6%) [93], Egypt (6.7%) [58], and Italy (1.4%) [61].

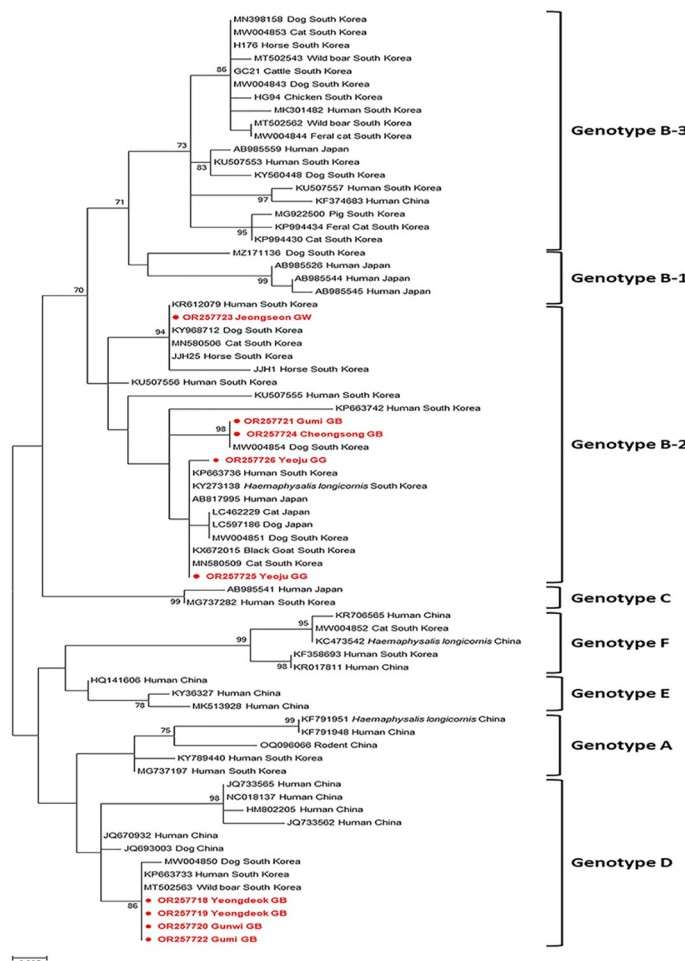

**Fig 7. Phylogenetic tree of the severe fever with thrombocytopenia syndrome virus based on the analysis of partial sequences of small segments.** Maximum-likelihood analysis was used to construct the Kimura 2-parameter model (1,000 bootstrap replicates). The scale bar implies nucleotide substitution per site. Sequences identified from *A. agrarius* are indicated in filled circles.

These differences may be explained by the rodent species and samples used for detection. In prior studies, *C. burnetii* detection was performed using various sample types such as blood, spleen, liver, and fecal samples. Consequently, the liver and spleen are considered suitable for the identification of *C. burnetii*. According to a previous study, the infection rate of *C. burnetii* in domestic livestock ranged from 6% to 22.7% depending on the species [57]. Although *C. burnetii* is a tick-borne pathogen, there are only a few reports of *C. burnetii* in ticks in the ROK [108,109]. Recent studies have reported co-infections of *C. burnetii* and SFTSV in ticks and humans [110, 111]; however, there was no co-infection with these two pathogens identified in rodents in this study. Despite the small number of *R. norvegicus* captured, *C. burnetii* infection was mostly detected in *R. norvegicus*; this could be because *R. norvegicus* may also serve as a reservoir in the ROK. A phylogenetic analysis based on the *IS1111* gene revealed the presence of genetic variations within the sequences identified in *A. agrarius*. One sequence formed the same clade with the virulent strains reported in Brazil, whereas the other exhibited high similarity to strains reported in different countries. The disadvantage of the *IS1111* gene is that it does not provide exact information, such as pathogenicity and species specificity (Fig 5); thus,

we cannot draw any conclusions about what the separate groupings within these *C. burnetii* sequences might represent. Further research is necessary to determine the pathogenicity of *C. burnetii* circulating in the ROK. The results obtained in the present study suggest that *A. agrarius* plays a role in the transmission of *C. burnetii* in humans and animals.

*Leptospira interrogans* is a rodent-borne pathogen, and accordingly, it was the third most frequently detected (15.4%) in this study. Our results demonstrated a relatively high prevalence compared with that reported in previous studies [88,112]. This is the first time that *Leptospira* spp. have been investigated in rodents through sampling of extensive regions in the ROK. Compared with those reported in other countries, the infection rates ranged from 1.3% to 35.2% [113–117]. *R. norvegicus* is also an important reservoir of this pathogen [72]; however, *L. interrogans* was detected in only one *R. norvegicus* and mostly detected in *A. agrarius*. Considering that *R. norvegicus* is commonly found around barns and farmhouses, this species also plays a critical role in the transmission of leptospirosis in domestic animals and humans. To date, *L. interrogans* has been divided into 23 serogroups based on serological methods and subdivided into more than 300 serovars [72]. The serovars circulating in each country are different, but the most frequently reported serovar worldwide is *Icterohaemorrhagiae* [72]. In the ROK, only a few studies have been conducted on serovar *Lai* [88, 118]. The phylogenetic analysis revealed that of the three sequences from rodents, two were classified as serovar *Lai* and one as serovar *Manila*, consistent with the findings from a previous study [88]. Consequently, *Lai* and *Manilae* are considered epidemic serovars in the ROK. The most important limitations of this study are that a serological analysis such as a microscopic agglutination test was not performed and that only a small number of samples were sequenced; therefore, serovar comparisons by province were not conducted. Although the number of the sequenced samples was low, the *rpoB* gene used in this study can be applicable for detection and serovar identification of *L. interrogans*. Furthermore, for accurate identification of *L. interrogans* serovars, serological testing along with the PCR method is absolutely necessary. These results suggest that the existence of various serovars in each province cannot be ruled out.

Since its first identification in China, SFTSV has been primarily detected in Asia [74–78]. Due to its associated high mortality rate, there is significant interest in SFTSV [74,83,84]. In the present study, the infection rate of SFTSV in *A. agrarius* was 5.8%, and this is the first report to describe SFTSV infection in *A. agrarius* in the ROK. Our results were significantly lower than those reported in China (32.3%) [119]. Compared with the infection rates reported in other animals in the ROK, the prevalence of SFTSV in rodents was similar to that in wild boars (5.2%) [120] and ticks (6.0%) [121], but higher than that in cats (4.0%) [122], dogs (2.9%) [123], pigs (1.7%) [124], black goats (2.4%) [125], and wild animals (3.3%) [126]. However, the prevalence of SFTSV was the highest in feral cats (17.5%) in the ROK [127]. There has been a recent increase in the populations of feral cats, and they share habitats with wildlife, domestic animals, and humans. Several studies have demonstrated that SFTSV is transmitted to humans through direct contact with cats [128,129], suggesting that feral cats are infected by rodents. It is believed that SFTSV circulates in a zoonotic cycle between ticks and vertebrates [130]. Rodents are considered the representative reservoirs in maintaining tick-borne pathogens and may play a vital role in the transmission of SFTSV. The sequences obtained from *A. agrarius* belonged to subgenotype B-2 and D genotype; these results revealed a similar distribution in both genotypes. Sequences belonging to subgenotype B-2 are the most prevalent and associated with the highest mortality rate (43.8%) in the ROK [131], whereas genotype D is primarily found in China. Four sequences belonging to genotype D were identical to those found in a human patient in the ROK, suggesting that this genotype is pathogenic. Different genotypes of SFTSV have been shown to trigger different clinical manifestations in a ferret model [130]; however, although the clinical manifestations have not been confirmed in rodents, they

may be pathogenic to humans. To date, SFTSV has been detected in diverse animals, but no conclusions can be drawn on how the virus is transmitted to these animals. The results of the present study provide a clue for elucidating the transmission route of SFTSV, suggesting the need to establish a continuous monitoring and surveillance systems to minimize a serious risk of SFTSV infection.

## Conclusions

Urbanization and climate change affect not only on humans but also wildlife. The most significant concern caused by these changes is that the probability of disease transmission through ecosystem destruction has been significantly increasing compared with that observed in the past. This study investigated the prevalence of zoonotic pathogens in rodent populations through a systematic epidemiological investigation. Although we did not screen for all rodent-borne pathogens, the results indicated that, at least, rodents may act as critical reservoirs for *A. phagocytophilum*, *Bartonella* spp., *B. afzelii*, *C. burnetii*, *L. interrogans*, and SFTSV in the ROK. Our findings also demonstrated that rodents harbor several pathogens, implying the possibility of simultaneous transmission to humans. Most importantly, except for SFTSV, the pathogens investigated in this study are commonly misdiagnosed or underdiagnosed in the ROK; thus, their importance is neglected. Our findings indicate that rodents pose a potential risk to public health. Overall, our study provides useful information on rodent-borne pathogens and underscores the urgent need for rapid diagnosis, prevention, and control strategies for zoonotic diseases.

## Supporting information

**S1 Table. Number of captured rodents from each region in each province.**
(DOCX)

## Acknowledgments

We are very grateful to Dong-Hun Jang, Eun-Mi Kim, and Seung-Uk Shin for their valuable help in sample collection. All authors read and approved the final version.

## Author Contributions

**Conceptualization:** Kyoung-Seong Choi, Joon-Seok Chae.

**Data curation:** Kyoung-Seong Choi, Sunwoo Hwang.

**Formal analysis:** Kyoung-Seong Choi, Sunwoo Hwang, Myung Cheol Kim, Hyung-Chul Cho, Yu-Jin Park, Min-Jeong Ji, Sun-Woo Han.

**Investigation:** Sunwoo Hwang, Myung Cheol Kim.

**Supervision:** Kyoung-Seong Choi, Joon-Seok Chae.

**Visualization:** Kyoung-Seong Choi, Sunwoo Hwang, Hyung-Chul Cho.

**Writing – original draft:** Kyoung-Seong Choi, Sunwoo Hwang, Joon-Seok Chae.

**Writing – review & editing:** Kyoung-Seong Choi, Sunwoo Hwang, Myung Cheol Kim, Hyung-Chul Cho, Yu-Jin Park, Min-Jeong Ji, Sun-Woo Han, Joon-Seok Chae.

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
