## [Decision Letter · Decision Letter 0]

14 Mar 2024

Dear Dr. Choi,

Thank you very much for submitting your manuscript "Molecular surveillance of zoonotic pathogens from wild rodents in the Republic of Korea" for consideration at PLOS Neglected Tropical Diseases. As with all papers reviewed by the journal, your manuscript was reviewed by members of the editorial board and by several independent reviewers. In light of the reviews (below this email), we would like to invite the resubmission of a significantly-revised version that takes into account the reviewers' comments. 

We cannot make any decision about publication until we have seen the revised manuscript and your response to the reviewers' comments. Your revised manuscript is also likely to be sent to reviewers for further evaluation.

Sincerely,

Colleen B Jonsson, PhD

Academic Editor

Amy Morrison

Section Editor

Reviewer's Responses to Questions

**Key Review Criteria Required for Acceptance?**

**Methods**

-Are the objectives of the study clearly articulated with a clear testable hypothesis stated?

-Is the study design appropriate to address the stated objectives?

-Is the population clearly described and appropriate for the hypothesis being tested?

-Is the sample size sufficient to ensure adequate power to address the hypothesis being tested?

-Were correct statistical analysis used to support conclusions?

-Are there concerns about ethical or regulatory requirements being met?

Reviewer #1: There is some basic information that should be included in the methods. 

1. the general description of “near rivers, valleys, farms, mountains, and lakes” is ambiguous. Were any other data collected (specific habitats, environmental conditions, etc)? Perhaps the authors could use more helpful descriptors (e.g., periurban? rural? near urban centers?) This would help their discussion, where they conclude that “rodents were trapped from areas with frequent movement of people, which may be associated with a high probability of disease transmission…” They state that the sites were selected by SFTS cases, but should provide more information to help understand what/where exactly they sampled, ecologically.

2. The authors must state how the rodents were identified. Defining how the rodents were identified becomes an issue as ~11% of the captured rodents were “unknown” species. Were the authors simply more confident in their identification of (the very distinctive) Apodemus agrarius and Rattus norvegicus, but were less confident to identify other rodents? If they identified A. agrarius as “rodent with a stripe” then they should state that as the identification method - how did they know it was not a Sicista sp? How did they discriminate between Rattus rattus and Rattus norvegicus? 

3. The authors have to be more specific about their geographic scale. They refer to “province” in Tables 2 and 3, but display subdivisions of provinces in Figure 1. In the text they refer to “regions” and it seems region = province, but it is not clear. This should be very clearly defined, specifically stating the scale at which the data were analyzed, and explaining why the map shows a different scale than how the data were analyzed. For example, there appear to be 9 subdivisions of Gyeongbuk for which presence/absence is noted – are these the “regions” referred to in the text? Are these all regions that were sampled? Other examples: L276 – “Four regions” but 7 points are shown on Figure 1; L286 – “Five different regions” but 11 points are shown on Figure 1; L309 – “four different regions” but 6 points are shown on Figure 1; L330 – “found in all regions”. 

4. One has to consider trapping effort for “prevalence” to be meaningful. The authors wish to make conclusions comparing the observed infection ratios between pathogens/regions/species and make comparisons to historical data (e.g., L226-L231, L232-L235 and Table 3). The probability of detecting something increases with increased sampling. For example, I would argue that there is very little that they can conclude about the vector/reservoir status of R. norvegicus based on 5 captures. So, in addition to the above mentioned points (describing the sampling sites and clarifying the geographic scale), the authors must state and/or consider the sampling effort (trap-nights per sampling unit, e.g., per site? region? province?) when analyzing their data. Otherwise they must limit their presentation, analysis, and discussion to presence/absence in the ROK.

5. If the authors define their sampling strategy and state their sampling effort with respect to geographic scale, they have to include correct statistical comparisons to make any inferences. For example, L259 – “The most” is misleading. Even though Bartonella were detected in 48% of A. agrarius, the confidence interval for the estimated number of R. norvegicus expected to be Bartonella-positive is ~1-70% based on 1/5 captures, i.e., there is no difference in infection ratio.

Reviewer #2: -Are the objectives of the study clearly articulated with a clear testable hypothesis stated? Yes

-Is the study design appropriate to address the stated objectives? Yes

-Is the population clearly described and appropriate for the hypothesis being tested? Yes

-Is the sample size sufficient to ensure adequate power to address the hypothesis being tested? Yes

-Were correct statistical analysis used to support conclusions? N/A

-Are there concerns about ethical or regulatory requirements being met? Yes

Reviewer comments were added to uploaded document.

Reviewer #3: - **Objectives and Hypothesis**: The objectives of the study are outlined but could be more explicitly connected to a clear, testable hypothesis. Clarifying the hypothesis would enhance the study's focus and provide a clearer framework for evaluation.

- **Study Design**: The design, involving molecular surveillance across various regions, appears appropriate for addressing the broad objectives related to pathogen prevalence. However, the justification for specific methodological choices (e.g., nested PCR) need to be strengthened.

- **Population Description**: The rodent population is clearly described, and the choice of species seems appropriate for the study's focus on zoonotic pathogens. The geographical and ecological breadth covered is a strength. Although the sample size is substantial for a surveillance study of this nature, a power analysis to justify the sample size in relation to expected pathogen prevalence is missing and would enhance the methodology.

- **Statistical Analysis**: There is minimal discussion on the statistical methods used for analyzing the data. Expanding on this, especially regarding the handling of co-infections and multiple comparisons, would support the conclusions drawn.

**Results**

-Does the analysis presented match the analysis plan?

-Are the results clearly and completely presented?

-Are the figures (Tables, Images) of sufficient quality for clarity?

Reviewer #1: Table 3 – I suggest removing R. norvegicus from this table entirely. The status of the five specimens can be easily stated in the text. I think it is important to know which of the detections per province were from each species, and with 151 individuals, a Table is the best way to display the data for A. agrarius, but the R. norvegicus data makes these numbers ambiguous.

Reviewer #2: -Does the analysis presented match the analysis plan? Yes

-Are the results clearly and completely presented? No

-Are the figures (Tables, Images) of sufficient quality for clarity? Yes

Reviewer comments were added to uploaded document.

Reviewer #3: - **Clarity and Completeness**: The results are presented with significant findings regarding pathogen prevalence and co-infections. However, detailing in which tissues each pathogen was detected would provide a significant improvement.

- **Quality of Figures and Tables**: The provided figures and tables are adequate but could be enhanced for clarity, particularly regarding the visualization of co-infection patterns and their distribution across sampled regions.

**Conclusions**

-Are the conclusions supported by the data presented?

-Are the limitations of analysis clearly described?

-Do the authors discuss how these data can be helpful to advance our understanding of the topic under study?

-Is public health relevance addressed?

Reviewer #1: The authors make conclusions concerning relative prevalence/occurrence of the rodents and pathogens with respect to geography. Without a more rigorous sampling strategy, or at least a better described/defined sampling scheme, they are limited in making some of these comparisons. In addition, the discussion includes details that are not completely relevant to the study design (as I have understood it). They should focus on presence/prevalence. They have no data on pathogenicity, so I recommend removing these points. I have included an edited pdf copy of the manuscript with my suggestions of things to delete for their consideration, and they could probably remove more.

The study has an additional limitation that should absolutely be addressed. The authors chose to use PCR or conventional RT-PCR for identification, yet failed to generate sequences from a proportion of the “positives”. The authors should address this specifically in the discussion: is this an acceptable ratio in similar studies? Should these be considered “putative” positive? Why weren’t more sensitive and specific methods used (e.g., RTqPCR)? Could this also influence the ratio when comparing to other, previously published molecular surveys (e.g., Kim et al. 2022 Parasite & Vectors 15:486, doi: 10.1186/s13071-022-05608-w)?

Another minor limitation that the authors should address is that they provide no information about the structure of the cohort (males/females, adult/juvenile, etc). This is not completely necessary to include in the analysis, but should be mentioned/considered in the discussion as it may affect the observed prevalence (infection ratio); and I suspect it had a very large affect on the trapping success.

Reviewer #2: -Are the conclusions supported by the data presented? Yes

-Are the limitations of analysis clearly described? Yes

-Do the authors discuss how these data can be helpful to advance our understanding of the topic under study? Yes

-Is public health relevance addressed? Yes

Reviewer comments were added to uploaded document.

Reviewer #3: - **Support by Data**: The conclusions drawn about the prevalence and public health significance of the detected pathogens are generally supported by the data, albeit with the caveat that the methodological limitations could impact these findings.

- **Limitations**: The discussion of limitations, especially regarding methodological choices (nested PCR, partial genome sequencing), is needed. Acknowledging these could strengthen the study's credibility.

- **Advancement of Understanding**: The manuscript makes a valuable contribution by mapping zoonotic pathogen prevalence in rodents in the ROK.

**Editorial and Data Presentation Modifications?**

Reviewer #1: (No Response)

Reviewer #2: Reviewer comments were added to uploaded document.

Reviewer #3: (No Response)

**Summary and General Comments**

Reviewer #1: The authors screen for the presence of some bacterial pathogens and one arbovirus in 2 rodent species captured at targeted sites, based on human cases of the arbovirus. The main goal was to characterize the presence of these zoonotic pathogens in the Republic of Korea in two rodent species. They screen by molecular diagnostic methods, and report the infection ratios by species and by geographic unit. They also included some phylogenetic analyses of the detected pathogens, including some (but not all) of the "positives". Overall the methods require a bit more clarification, because I think the manuscript needs a better perspective on what can/not be inferred from the study design and data. As explained in other comments above, the geographic unit and sampling effort need to be clearly defined if the authors wish to infer prevalence based on geography. The phylogenetic analysis is fine, but the authors should limit the conclusions they can make from these data concerning pathogenicity. Simply stated, this could be a simple epidemiological survey for multiple pathogens. However, the authors could make some additional inferences (e.g., geographic distribution) if they are more scientific about the approach. Therefore, the discussion needs major revision, and that will depend on if/how the authors improve the methods.

In addition to the major points mentioned above, here are some minor suggestions / clarifications that would help the mansucript:

L135-L136: SFTSV was only briefly Huaiyangshan banyangvirus, has been Dabie bandavirus since 2018, and is now Bandavirus dabieense according to ICTV

L147 – I would rephrase this as “…rodents may be involved in the transmission cycles of various diseases.” Particularly as this is the hypothesis they intend to investigate.

L153 “…to investigate the occurrence of some rodent-borne diseases…”

L222 – I am confused by this sentence. I was under the impression that the authors only captured rodents in the ROK, but here they state that A. agrarius was captured “mostly” in ROK (and also somewhere else?). Perhaps the authors mean that A. agrarius was found at all sites (?) but R. norvegicus was only captured at 2 sites? Two regions? 

L172 – define “serum separating tube (SST)”

L267 – “…formed the same group as B. grahamii found in leeches (KX270012) and found in A. agrarius (JN810851) reported in the ROK…”

L279 and Figure 1 – B. burgdorferi is listed in the figure but the phylogenetic analysis in Figure 4 and text L279 state that this is B. afzelii. Please clarify in the text or write “B. burgdorferi s.l.” on the figure.

L292-293 what was 

---

## [Decision Letter · Decision Letter 1]

13 May 2024

Dear Dr. Choi,

Thank you very much for submitting your manuscript "Molecular surveillance of zoonotic pathogens from wild rodents in the Republic of Korea" for consideration at PLOS Neglected Tropical Diseases. As with all papers reviewed by the journal, your manuscript was reviewed by members of the editorial board and by several independent reviewers. The reviewers appreciated the attention to an important topic. Based on the reviews, we are likely to accept this manuscript for publication, providing that you modify the manuscript according to the review recommendations. 

Sincerely,

Colleen B Jonsson, PhD

Academic Editor

Amy Morrison

Section Editor

Reviewer's Responses to Questions

**Key Review Criteria Required for Acceptance?**

**Methods**

-Are the objectives of the study clearly articulated with a clear testable hypothesis stated?

-Is the study design appropriate to address the stated objectives?

-Is the population clearly described and appropriate for the hypothesis being tested?

-Is the sample size sufficient to ensure adequate power to address the hypothesis being tested?

-Were correct statistical analysis used to support conclusions?

-Are there concerns about ethical or regulatory requirements being met?

Reviewer #1: The authors have improved the methods description by now including a table with regions sampled within provinces, as well as defining ecological categories that describe each region. They added more information, but there the sampling effort is still ambiguous. The authors must be very clear about trap effort. Per the methods, they define 1 trap-night = 60 traps. *State how many trap-nights were performed in each region.*

Reviewer #2: Line 173-178 – Please clarify, was each capture location only sampled once or were they sampled multiple times?

**Results**

-Does the analysis presented match the analysis plan?

-Are the results clearly and completely presented?

-Are the figures (Tables, Images) of sufficient quality for clarity?

Reviewer #1: L316 (previously L293) The authors have included bootstrap values on the phylogenetic trees. However, they did not answer my previous question concerning the use of the word "clades". How were the clades defined? The authors state “there are several clades” – how did they define these clades? There do not seem to be any “clades” in Figure 5 and I wonder why/how one could describe genetic variability (or define clades) with a 203 bp segment. The best solution would be to remove the mention of “clades”. There are specific methods for defining genetic clusters (not necessarily high bootstrap support), and those methods were not used in the manuscript. This is a minor point, and it is only a suggestion for improvement, as many people write “clades” or “clusters” without such formal analysis. I was confused about their responses about this, so I thought I would mention it again.

Reviewer #2: Figure 6 – Please state what the box of dash line indicates.

Line 370 – Please add citation for statement about difference in sensitivity by nested-PCR and qPCR.

Line 373 – Please elaborate what is meant by “sequences were poor.”

**Conclusions**

-Are the conclusions supported by the data presented?

-Are the limitations of analysis clearly described?

-Do the authors discuss how these data can be helpful to advance our understanding of the topic under study?

-Is public health relevance addressed?

Reviewer #1: The authors have not addressed the issue of PCR-positive/negative. They have included information about how many sequences were “selected” for analysis (209-210), but that is clearly not true, nor is it helpful resolving the issue raised previously. For example, clearly all 15 “successfully” sequenced A. phagocytophilum were included in Figure 2, and not “<5” as stated in the methods. Regardless, the issue is whether “unsuccessful” sequencing of a PCR product means that the sample is negative. The authors attempted to address these limitations in the discussion (paragraph beginning L369), but they have not addressed the issue of false-positivity. 

I disagree with their statement that nested PCR is more sensitive and specific than qPCR (L369), and I would counter that nested PCR is more prone to false-positive results, and therefore less specific. The classical way to confirm conventional PCR (positivity and specificity) is to sequence the resulting amplicons. Otherwise, the samples should be considered putatively/potentially positive. The authors clearly state that 15 of 21 Anaplasma-positive (L269), 15 of 78 Bartonella-positives (L284), 2 of 5 Borellia-positives, etc were *successfully* sequenced. That is a significant proportion of PCR-positives. They mentioned that sequencing was not successful in the discussion (L431, L483). That is a huge "red flag" that those were not true positives.

The explanations the authors provide in the discussion (a “low density infection” or the fact that the primers do not match some Leptospira serovars) are unclear and create some additional problems. 

The way to resolve this, as I suggested last time as well, is for the authors to state in the discussion that they considered PCR-positives to be truly positive, regardless of whether the amplicon could be sequenced, and note that this may overestimate the number of true positives, as some amplicons could not be sequenced. I suggest that they do not speculate that it is due to a “low density infection” (L431), and they do not imply that their PCR was inefficient (L483).

Reviewer #2: (No Response)

**Editorial and Data Presentation Modifications?**

Reviewer #1: L136 It is worth revising this more precisely. The virus is still called “severe fever with thrombocytopenia syndrome virus” and should still be called that. However, the official scientific name is Bandavirus dabieense. Just like the “striped field mouse” is Apodemus agrarius.

L255 correct the spelling of SFTSV

Reviewer #2: (No Response)

**Summary and General Comments**

Reviewer #1: The authors have made many improvements to the manuscript. I think there are still a few minor points that should be corrected. 

There are also some changes in the revision that I disagree with. I do not agree with the complete removal of information regarding unidentified rodents. It makes sense to concentrate the study on one species for the manuscript. However, I would expect that one would collect a variety of species using their stated trapping method. It now seems as though the authors only captured two species. This is suspicious and unlikely. 

I do not agree with completely removing the pathogen-detection results from R norvegicus. My previous comments were intended to help clarify the data presentation and analysis, I was not suggesting the authors censor their data. Now it seems strange that the manuscript refers to pathogens in “rodents”, when, actually, the authors only present data from A agrarius. It is also strange that they state only 156 rodents were captured (which is untrue), list two species that they could identify, and then remove R norvegicus from the analysis.

There are two alternatives, as I see it: they state how many rodents were captured in total (175), and then state that this study focused on identifying human pathogens in the 151 A. agrarius (86%). Don't mention any other species at all. 

Alternatively, they should return the results to how it was written originally: "A total of 175 rodents were captured and morphologically classified as follows: Apodemus agrarius (striped field mouse) (n = 151), Rattus norvegicus (Norway rat) (n = 5), and unknown (n = 19)." This was completely fine. My original comment was that including all rodents in the table was confusing. As there were only 5 brown rats, they could have simply stated these results in the respective sections (e.g., "Bartonella was detected in 3 brown rats").

Reviewer #2: Please find below minor points and suggestions that may help. 

Line 42 – Is the author’s suggestion for the ROK or worldwide as it known that rodents harbor various pathogens?

Line 148-149 – Sentence construction and word choice

Line 269 – “indicating that this pathogen was spread throughout the ROK.” 

Line 345 – Suggest removing the word potentially as the pathogens characterized in the manuscript are zoonotic pathogens.

Line 407-410 – Please clarify sentence and the point the author is conveying.

Line 1020 – Sentence construction and suggest the use of the word symbol instead of markings.

PLOS authors have the option to publish the peer review history of their article (what does this mean?). If published, this will include your full peer review and any attached files.

Reviewer #1: No

Reviewer #2: No

Figure Files:

Data Requirements:

Reproducibility:

References

---

## [Editor Report · Decision Letter 2]

21 Jun 2024

Dear Dr. Choi,

We are pleased to inform you that your manuscript 'Molecular surveillance of zoonotic pathogens from wild rodents in the Republic of Korea' has been provisionally accepted for publication in PLOS Neglected Tropical Diseases.

Best regards,

Colleen B Jonsson, PhD

Academic Editor

Amy Morrison

Section Editor

---

## [Editor Report · Acceptance letter]

3 Jul 2024

Dear Dr. Choi,

We are delighted to inform you that your manuscript, "Molecular surveillance of zoonotic pathogens from wild rodents in the Republic of Korea," has been formally accepted for publication in PLOS Neglected Tropical Diseases.

Best regards,

Shaden Kamhawi

co-Editor-in-Chief

Paul Brindley

co-Editor-in-Chief
